# Multimodal Virtual Point 3D Detection

**Tianwei Yin**
UT Austin
yintianwei@utexas.edu

**Xingyi Zhou**
UT Austin
zhouxy@cs.utexas.edu

**Philipp Krähenbühl**
UT Austin
philkr@cs.utexas.edu

## Abstract

Lidar-based sensing drives current autonomous vehicles. Despite rapid progress, current Lidar sensors still lag two decades behind traditional color cameras in terms of resolution and cost. For autonomous driving, this means that large objects close to the sensors are easily visible, but far-away or small objects comprise only one measurement or two. This is an issue, especially when these objects turn out to be driving hazards. On the other hand, these same objects are clearly visible in onboard RGB sensors. In this work, we present an approach to seamlessly fuse RGB sensors into Lidar-based 3D recognition. Our approach takes a set of 2D detections to generate dense 3D virtual points to augment an otherwise sparse 3D point cloud. These virtual points naturally integrate into any standard Lidar-based 3D detectors along with regular Lidar measurements. The resulting multi-modal detector is simple and effective. Experimental results on the large-scale nuScenes dataset show that our framework improves a strong CenterPoint baseline by a significant 6.6 mAP, and outperforms competing fusion approaches. Code and more visualizations are available at https://tianweiy.github.io/mvp/.

## 1 Introduction

3D perception is a core component in safe autonomous driving [1, 55]. A 3D Lidar sensor provides accurate depth measurements of the surrounding environment [23, 49, 75], but is costly and has low resolution at long range. A top-of-the-line 64-lane Lidar sensor can easily cost more than a small car with an input resolution that is at least two orders of magnitude lower than a $50 RGB sensor. This Lidar sensor receives one or two measurements for small or far away objects, whereas a corresponding RGB sensor sees hundreds of pixels. However, the RGB sensor does not perceive the depth and cannot directly place its measurements into a scene.

In this paper, we present a simple and effective framework to fuse 3D Lidar and high-resolution color measurements. We lift RGB measurements into 3D virtual points by mapping them into the scene using close-by depth measurements of a Lidar sensor (See Figure 1 for an example). Our **M**ulti-modal **V**irtual **P**oint detector, **MVP**, generates high-resolution 3D point-cloud near target objects. A center-based 3D detector [66] then identifies all objects in the scene. Specifically, MVP uses 2D object detections to crop the original point cloud into instance frustums. MVP then generates dense 3D virtual points near these foreground points by lifting 2D pixels into 3D space. We use depth completion in image space to infer the depth of each virtual point. Finally, MVP combines virtual points with the original Lidar measurements as input to a standard center-based 3D detector [66].

Our multi-modal virtual point method has several key advantages: First, 2D object detections are well optimized [17, 74] and highly accurate even for small objects. See Figure 2 for a comparison of two state-of-the-art 2D and 3D detectors on the same scene. The 2D detector has a significantly higher 2D detection accuracy but lacks the necessary 3D information used in the downstream driving task. Secondly, virtual points reduce the density imbalance between close and faraway objects. MVP augments objects at different distances with the same number of virtual points, making the point cloud measurement of these objects more consistent. Finally, our framework is a plug-and-

35th Conference on Neural Information Processing Systems (NeurIPS 2021).

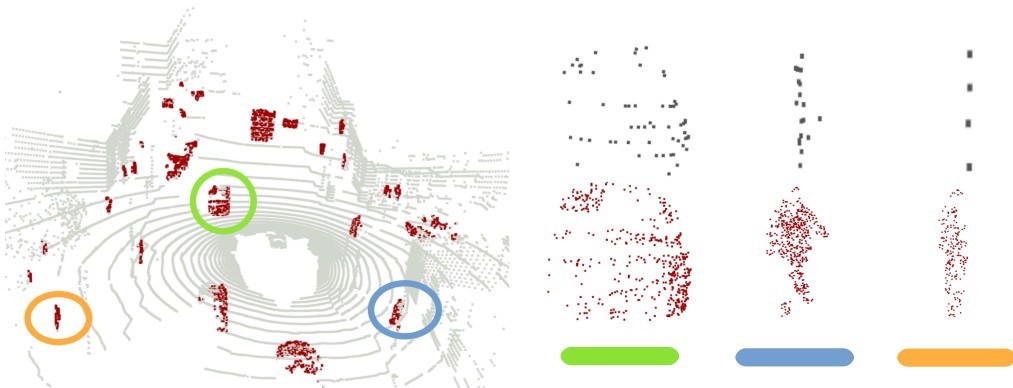

Figure 1: We augment sparse Lidar point cloud with dense semantic virtual points generated from 2D detections. Left: the augmented point-cloud in the scene. We show the original points in gray and augmented points in red. Right: three cutouts with the origial points on top and virtual points below. The virtual points are up to two orders of magnitude denser.

play module to any existing or new 2D or 3D detectors. We test our model on the large-scale nuScenes dataset [2]. Adding multi-modal virtual points brings **6.6 mAP** improvements over a strong CenterPoint baseline [66]. Without any ensembles or test-time augmentation, our best model achieves **66.4 mAP** and **70.5 NDS** on nuScenes, outperforming all competing non-ensembled methods on the nuScenes leaderboard at the time of submission.

## 2    Related work

**2D Object Detection**    has great progress in recent years. Standard approaches include the RCNN family [13, 17, 43] which first predict class-agnostic bounding boxes based on predefined anchor boxes and then classify and refine them in a two-stage fashion with deep neural networks. YOLO [42], SSD [33], and RetinaNet [30] predicts the class specific bounded boxes in one shot. Recent anchor-free detectors like CornerNet [24] and CenterNet [74] directly localize objects through keypoints without the need of predefined anchors. In our approach, we use CenterNet [74] as our 2D detector for its simplicity and superior performance for detecting small objects. See Figure 2 for an example of a 2D detectors output.

**Lidar-based 3D Object Detection**    estimates rotated 3D bounding boxes from 3D point clouds [7, 12, 23, 37, 60–63, 65, 76]. 3D detectors share a common output representation and network structure with 2D detectors but encode the input differently. VoxelNet [75] uses a PointNe-based feature extractor to generate a voxel-wise feature representation from which a backbone consisted of sparse 3D convolutions and bird-eye view 2D convolution produces detection outputs. SECOND [60] introduces more efficient sparse convolution operations. PIXOR [61] and PointPillars [23] directly process point clouds in bird-eye view, further improving efficiency. Two-stage 3D detectors [8, 45–47, 63] use a PointNet-based set abstraction layer [39] to aggregate RoI-specifc features inside first stage proposals to refine outputs. Anchor-free approaches [5, 36, 57, 59, 61, 66] remove the need for axis-aligned bird-eye view anchor boxes. VoteNet [36] detects 3D objects through Hough voting and clustering. CenterPoint [66] proposes a center-based representation for 3D object detection and tracking and achieved state-of-the-art performance on nuScenes and Waymo benchmarks. However, as Figure 2 shows a Lidar-only detector still misses small or far-away objects due to the sparsity of depth measurements. In this work, we build upon the CenterPoint detector and demonstrate significant **6.6 mAP** improvements by adding our multi-modal virtual point approach.

**Camera-based 3D Object Detection**    Camera-based 3D object detection predicts 3D bounding boxes from camera images. Mono3D [6] uses the ground-plane assumption to generate 3D candidate boxes and scores the proposals using 2D semantic cues. CenterNet [74] first detects 2D objects in images and predicts the corresponding 3D depth and bounding box attributes using center features. Despite rapid progress, monocular 3D object detectors still perform far behind the Lidar-based

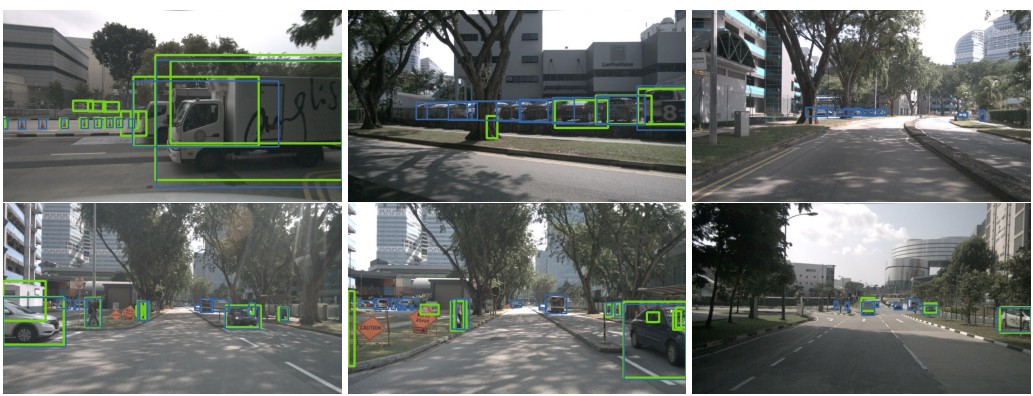

Figure 2: Comparison between state-of-the-art image-based 2D detector [73] and point cloud based 3D detector [66]. We show detection from the 2D detector in blue and detection from 3D detector in green. For the 3D detector, we project the predicted 3D bounding boxes into images to get the 2D detections. For the 2D detector, we train the model using projected 2D boxes from 3D annotations. Compared to 2D detector, 3D detector often misses faraway or small objects. A quantitative comparison between 2D and 3D detectors is included in Section 5.2.

methods. On state-of-the-art 3D detection benchmarks [2, 12], state-of-the-art monocular methods [26, 41] achieve about half the mAP detection accuracy, of standard Lidar based baselines [60]. Pseudo-Lidar [56] based methods produce a virtual point cloud from RGB images, similar to our approach. However, they rely on noisy stereo depth estimates [25, 40, 56] while we use more accurate Lidar measurements. Again, the performance of purely color-based approaches lags slightly behind Lidar or fusion-based methods [2, 12].

**Multi-modal 3D Object Detection**   fuses information of Lidar and color cameras [20, 28, 28, 29, 35, 37, 53, 67, 71, 72]. Frustum PointNet [38] and Frustum ConvNet [58] first detect objects in image space to identify regions of interest in the point cloud for further processing. It improves the efficiency and precision of 3D detection but is fundamentally limited by the quality of 2D detections. In contrast, we adopt a standard 3D backbone [75] to process the augmented Lidar point cloud, combining the benefit of both sensor modalities. MV3D [7] and AVOD [22] performs object-centric fusion in a two-stage framework. Objects are first detected in each sensor and fused at the proposal stage using RoIPooling [43]. Continuous fusion [20, 29] shares image and Lidar features between their backbones. Closest to our approach are MVX-Net [48], PointAugmenting [54], and PointPainting [52], which utilize point-wise correspondence to annotate each lidar point with image-based segmentation or CNN features. We instead augment the 3D lidar point cloud with additional points surrounding 3D measurements. These additional points make full use of the higher dimensional RGB measurements.

**Point Cloud Augmentation**   generates denser point clouds from sparse Lidar measurements. Lidar-based methods like PUNet [69], PUGAN [27], and Wang et al. [64] learn high level point-wise features from raw Lidar scans. They then reconstruct multiple upsampled point clouds from each high dimensional feature vector. Image-based methods [18, 21, 51] perform depth completion from sparse measurements. We build upon these depth completion methods and demonstrate state-of-the-art 3D detection results through point upsampling.

## 3   Preliminary

Our framework relies on both 2D detection, existing 3D detectors, and a mapping between 2D and 3D. We introduce the necessary concepts and notations below.

**2D Detection.**   Let $I$ be a camera image. A 2D object detector aims to localize and classify all objects in $I$. A bounding-box $b_i \in \mathbb{R}^4$ describes the objects location. A class score $s_i(c)$ predicts the likelihood of detection $b_i$ to be of class $c$. An optional instance mask $m_i \in [0, 1]^{W \times H}$ predicts a pixel-level segmentation of each object. In this paper, we use the popular CenterNet [74] detector. CenterNet detects objects through keypoint estimation. It takes the input image $I$ and predicts a

heatmap for each class $c$. Peaks (local maxima) of the heatmap corresponds to an object. The model regresses to other bounding box attributes using peak features with an L1 [74] or box IoU [44] objective. For instance segmentation, we use CenterNet2 [73] which adds a cascade RoI heads [3] on top of the first stage proposal network. The overall network runs at 40 FPS and achieves 43.3 instance segmentation mAP on the nuScenes image dataset [2].

**3D Detection.** Let $P = \{(x, y, z, r)_i\}$ be a point cloud with 3D location $(x, y, z)$ and reflectance $r$. The goal of a 3D detector is to predict a set of 3D bounding boxes $\{b_i\}$ from the point cloud $P$. The bounding box $b = (u, v, o, w, l, h, \theta)$ includes the 3D center location $(u, v, o)$, object size $(w, l, h)$ and the yaw rotation along z axis $\theta$. In this paper, we build upon the state-of-the-art CenterPoint [66] detector. We experiment with two popular 3D backbones: VoxelNet [75] and PointPillars [23]. VoxelNet quantizes the irregular point clouds into regular bins followed by a simple average pooling to extract features from all points inside a bin [60]. After that, a backbone consisted of sparse 3D convolutions [14] processes the quantized 3D feature volumes and the output is a map view feature map $M \in \mathbb{R}^{W \times H \times F}$. PointPillar directly processes point clouds as bird-eye view pillar, a single elongated voxel per map location, and extracts features with fast 2D convolution to get the map view feature map $M$.

With the map view features, a detection head inspired by CenterNet [74] localizes objects in bird-eye view and regress to other box parameters using center features.

**2D-3D Correspondence.** Multi-modal fusion approaches [48, 52, 53, 72] often rely on a point-wise correspondence between 3D point clouds and 2D pixels. In absence of calibration noise, the projection from the 3D Lidar coordinate into a 2D image coordinate involves an SE(3) transformation from the Lidar measurement to the camera frame and a perspective projection from the camera frame into image coordinates. All transformations may be described with homogeneous, time-dependent transformations. Let $t_1$ and $t_2$ be the capture time of the Lidar measurement and RGB image respectively. Let $T_{(\text{car}\leftarrow\text{lidar})}$ be the transformation from the Lidar sensor to the reference frame of the car. Let $T_{(t_1 \leftarrow t_2)}$ be the transformation of the car between $t_2$ and $t_1$. Let $T_{(\text{rgb}\leftarrow\text{car})}$ be the transformation from the cars reference frame to the RGB sensor. Finally, let $P_{\text{rgb}}$ be the projection matrix of the RGB camera defined by the camera intrinsic. The transformation from the Lidar to RGB sensor is then defined by

$$T_{\text{rgb}\leftarrow\text{lidar}}^{t_1 \leftarrow t_2} = T_{(\text{rgb}\leftarrow car)} T_{(t_1 \leftarrow t_2)} T_{(\text{car}\leftarrow\text{lidar})}, \tag{1}$$

followed by a perspective projection with camear matrix $P_{\text{rgb}}$ and a perspective division. The perspective division makes the mapping from Lidar to RGB surjective and non-invertible. In the next section, we show how to recover an inverse mapping by using depth measurements of Lidar when mapping RGB to Lidar.

## 4  Multimodal Virtual Point

Given a set of 2D object detections, we want to generate dense virtual points $v_i = (x, y, z, \mathbf{e})$ where $(x, y, z)$ is the 3D location and $\mathbf{e}$ is the semantic feature from the 2D detector. For simplicity, we use the 2D detectors class scores as semantic features. For each detection $b_j$ with associated instance mask $\mathbf{m}_j$ we generate a fixed number $\tau$ multimodal virtual points.

**Virtual Point Generation.** We start by projecting the 3D Lidar point cloud onto our detection. Specifically, we transform each Lidar point $(x, y, z, r)_i$ into the reference frame of the RGB camera following Equation (1), then project it into image coordinates $\mathbf{p}_i$ with associated depth $d_i$ using a perspective projection. Let the collection of all projected points and depth values for a single detection $j$ be the objects frustum $\mathbf{F}_j = \{(\mathbf{p}_i, d_i) | \mathbf{p}_i \in \mathbf{m}_j \forall_i\}$. The frustum only considers projected 3D points $\mathbf{p}_i$ that fall within a detection mask $\mathbf{m}_j$. Any Lidar measurement outside detection masks is discarded. Next, we generate virtual points from each frustum $\mathbf{F}_j$.

We start by randomly sampling 2D points $\mathbf{s} \in \mathbf{m}$ from each instance mask $\mathbf{m}$. We sample $\tau$ points uniformly at random without repetition. For each sampled point $\mathbf{s}_k$, we retrieve a depth estimate $d_k$ from its nearest neighbor in the frustum $F_j$: $d_k = \arg\min_{d_i} \|\mathbf{p}_i - \mathbf{s}_k\|$. Given the depth estimate, we unproject the point back into 3D and append the object's semantic feature $e_j$ to the virtual point. We concatenate the one-hot encoding of the detected class and the detections objectness score in the semantic feature.

**Algorithm 1:** Multi-modal Virtual Point Generation

---

**Input**            : Lidar point cloud $\mathbf{L} = \{(x, y, z, r)_i\}$.

                       Instance masks $\{\mathbf{m}_1, \ldots, \mathbf{m}_n\}$ for $n$ objects.

                       Semantic features $\{\mathbf{e}_1, \ldots, \mathbf{e}_n\}$ with $\mathbf{e}_j \in \mathbb{R}^D$

                       Transformation $T_{\mathrm{rgb}\leftarrow\mathrm{lidar}}^{t_1\leftarrow t_2} \in \mathbb{R}^{4\times4}$ and camera projection $P_{\mathrm{rgb}} \in \mathbb{R}^{4\times4}$.

**Hyper parameters :** Number of virtual points per object $\tau$

**Output**          : Multi-modal 3D virtual points $\mathbf{V} \in \mathbb{R}^{n\times\tau\times(3+D)}$

$\mathbf{F}_j \leftarrow \emptyset \quad \forall_{j\in\{1\ldots n\}}$;                          `// Point cloud instance frustums`

**for** $(x_i, y_i, z_i, r_i) \in \mathbf{L}$ **do**

     `/* Perspective projection to 2D point `$\mathbf{p}$` depth `$d$` */`

     $\mathbf{p}, d \leftarrow Project\left(P_{\mathrm{rgb}}T_{\mathrm{rgb}\leftarrow\mathrm{lidar}}^{t_1\leftarrow t_2}(x_i, y_i, z_i, 1)^\top\right)$;

     **for** $j \in \{1 \ldots n\}$ **do**

         **if** $\mathbf{p} \in \mathbf{m}_j$ **then**

             $\mathbf{F}_j \leftarrow \mathbf{F}_j \cup \{(\mathbf{p}, d)\}$;                 `// Add point to frustum`

         **end**

     **end**

**end**

**for** $j \in \{1 \ldots n\}$ **do**

     $\mathbf{S} \leftarrow Sample_\tau(\mathbf{m}_j)$;       `// Uniformly sample `$\tau$` 2d points in instance mask`

     **for** $\mathbf{s} \in \mathbf{S}$ **do**

         $(\mathbf{p}, d) \leftarrow NN(\mathbf{s}, \mathbf{F}_j)$;            `// Find closest projected 3D point`

         `/* Unproject the 2D point `$\mathbf{s}$` using the nearest neighbors depth `$d$` */`

         $\mathbf{q} \leftarrow \left(P_{\mathrm{rgb}}T_{\mathrm{rgb}\leftarrow\mathrm{lidar}}^{t_1\leftarrow t_2}\right)^{-1} Unproject(\mathbf{s}, d)$;

         Add $(\mathbf{q}, \mathbf{e}_j)$ to $\mathbf{V}_j$;

     **end**

**end**

---

The virtual point generation is summarized in Algorithm 1 and Figure 3. Next, we show how to incorporate virtual points into a point-based 3D detector.

**Virtual Point 3D detection.** Voxel-based 3D detectors [60, 66] first voxelize 3D points $(x, y, z)_i$ and average all point features $(x, y, z, t, r)_i$ within a voxel. Here $r_i$ is a reflectance measure and $t_i$ is the capture time. A standard 3D convolutional network uses these voxelized features in further processing. For virtual points, this creates an issue. The feature dimensions of real points $(x, y, z, t, r)$ and virtual points differ $(x, y, z, t, \mathbf{e})$. A simple solution could be to either concatenate virtual and real points into a larger feature $(x, y, z, t, r, \mathbf{e})$ and set any missing information to zero. However, this is both wasteful, as the dimension of real points grows by $3\times$, and it creates an imbalanced ratio between virtual and real points in different parts of the scene. Furthermore, real measurements are often a bit more precise than virtual points and simple averaging of the two blurs out the information contained in real measurements. To solve this, we modify the average pooling approach by separately averaging features of virtual and real points and concatenating the final averaged features together as input to 3D convolution. For the rest of the architecture, we follow CenterPoint [66].

We further use virtual points in a second-stage refinement. Our MVP model generates dense virtual points near target objects which help two-stage refinement [45, 66]. Here, we follow Yin *et al.* [66] to extract bird-eye view features from all outward surfaces of the predicted 3D box. The main difference to Yin *et al.* is that our input is much denser around objects, and hence the second stage refinement has access to richer information.

## 5 Experiments

We evaluate our proposed multimodal virtual point method on the challenging nuScenes benchmark.

**nuScenes** [2] is a popular multimodal autonomous driving datasets for 3D object detection in urban scenes. The dataset contains 1000 driving sequences, with each 20s long and annotated with 3D

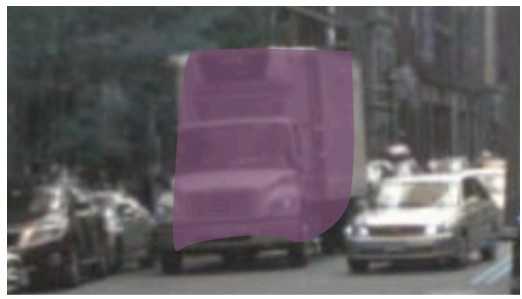

(a) 2D instance segmentation

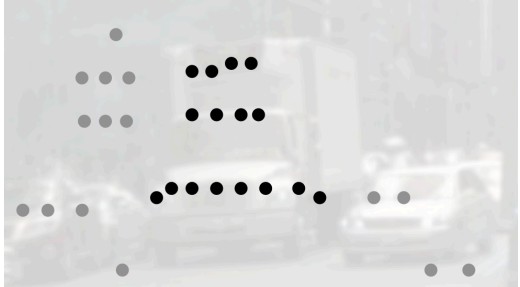

(b) Lidar point cloud projection

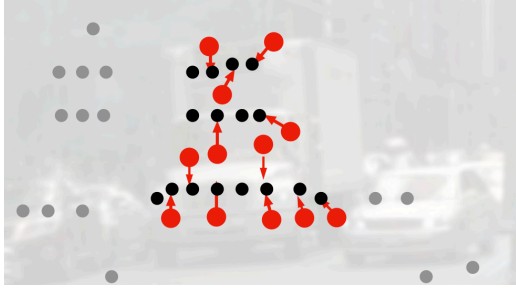

(c) Sampling and nearest neighbor matching

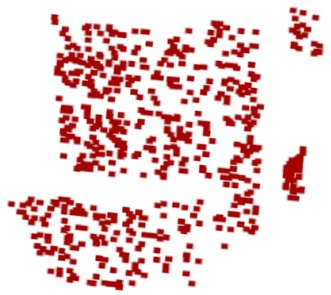

(d) Reprojected virtual points

Figure 3: Overview of our mlutimodal virtual point generation framework. We start by extracting 2D instance masks for each object in a color image (a). We then project all Lidar measurements into the reference frame of the RGB camera (b). For visualization purposes, points inside the objects are black, other points are grey. We then sample random points inside each 2D instance mask and retrieve a depth estimate from their nearest neighbor Lidar projection (c). For visualization clarity, (c) only shows a subset of virtual points. Finally, all virtual points are reprojected into the original point-cloud (d).

bounding boxes. The Lidar frequency is 20Hz and the dataset provides sensor and vehicle pose information for each Lidar frame but only includes object annotation every ten frames (0.5s). The dataset hides any personally identifiable information, blurs faces and license plates in color images. There are in total 6 RGB cameras at a resolution of $1600 \times 900$ and a capture frequency of 12Hz. We follow the official dataset split to use 700, 150, 150 sequences for training, validation, and testing. This in total results in 28130 frames for training, 6019 frames for validation, and 6008 frames for testing. The annotations include a fine-grained label space of ten classes with a long-tail distribution. For 3D object detection, the official evaluation metrics include the mean Average Precision (mAP)[11] and nuScenes detection score (NDS) [2]. mAP measures the localization precision using a threshold based on the birds-eye view center distance < 0.5m, 1m, 2m, 4m. NDS is a weighted combination of mAP and regression accuracy of other object attributes including box size, orientation, translation, and class-specific attributes [2]. NDS is the main ranking metric for the benchmark.

**Implementation Details.** Our implementation is based on the opensourced code of Center-Point [1] [66] for 3D detection and CenterNet2 [2][73] for 2D Detection.

For 2D detection, we train a CenterNet [74] detector on the nuScenes image dataset [2]. We use the DLA-34 [68] backbone with deformable convolutions [10]. We add cascade RoI heads [3] for instance segmentation following Zhou et al. [73]. We train the detector on the nuScenes dataset using the SGD optimizer with a batch size of 16 and a learning rate of 0.02 for 90000 iterations.

For 3D detection, we use the same VoxelNet [75] and PointPillars [23] architectures following [23, 66, 76]. For VoxelNet, the detection range is $[-54m, 54m]$ for the $X, Y$ axis and $[-5m, 3m]$ for the $Z$ axis while the range is $[-51.2m, 51.2m]$ for the $X, Y$ axis for the PointPillar architecture. The voxel

---

[1] https://github.com/tianweiy/CenterPoint
[2] https://github.com/xingyizhou/CenterNet2

Table 1: Comparisons with previous methods on nuScenes test set. We show the NDS, mAP, and mAP for each class. Abbreviations are construction vehicle (CV), pedestrian (Ped), motorcycle (Motor), and traffic cone (TC).

| Method | mAP | NDS | Car | Truck | Bus | Trailer | CV | Ped | Motor | Bicycle | TC | Barrier |
|---|---|---|---|---|---|---|---|---|---|---|---|---|
| PointPillars [23] | 30.5 | 45.3 | 68.4 | 23.0 | 28.2 | 23.4 | 4.1 | 59.7 | 27.4 | 1.1 | 30.8 | 38.9 |
| WYSIWYG [19] | 35.0 | 41.9 | 79.1 | 30.4 | 46.6 | 40.1 | 7.1 | 65.0 | 18.2 | 0.1 | 28.8 | 34.7 |
| 3DSSD [62] | 42.6 | 56.4 | 81.2 | 47.2 | 61.4 | 30.5 | 12.6 | 70.2 | 36.0 | 8.6 | 31.1 | 47.9 |
| PMPNet [65] | 45.4 | 53.1 | 79.7 | 33.6 | 47.1 | 43.1 | 18.1 | 76.5 | 40.7 | 7.9 | 58.8 | 48.8 |
| PointPainting [52] | 46.4 | 58.1 | 77.9 | 35.8 | 36.2 | 37.3 | 15.8 | 73.3 | 41.5 | 24.1 | 62.4 | 60.2 |
| CBGS [76] | 52.8 | 63.3 | 81.1 | 48.5 | 54.9 | 42.9 | 10.5 | 80.1 | 51.5 | 22.3 | 70.9 | 65.7 |
| CVCNet [4] | 55.3 | 64.4 | 82.7 | 46.1 | 46.6 | 49.4 | 22.6 | 79.8 | 59.1 | 31.4 | 65.6 | 69.6 |
| HotSpotNet [5] | 59.3 | 66.0 | 83.1 | 50.9 | 56.4 | 53.3 | 23.0 | 81.3 | 63.5 | 36.6 | 73.0 | 71.6 |
| CenterPoint [66] | 58.0 | 65.5 | 84.6 | 51.0 | 60.2 | 53.2 | 17.5 | 83.4 | 53.7 | 28.7 | 76.7 | 70.9 |
| **MVP (Ours)** | **66.4** | **70.5** | **86.8** | **58.5** | **67.4** | **57.3** | **26.1** | **89.1** | **70.0** | **49.3** | **85.0** | **74.8** |

size is $(0.075m, 0.075m, 0.2m)$ and $(0.2m, 0.2m, 8m)$ for VoxelNet and PointPillar respectively. For data augmentation, we follow CenterPoint and use global random rotations between $[-\pi/4, \pi/4]$, global random scaling between $[0.9, 1.1]$ and global translations between $[-0.5m, 0.5m]$. To deal with the long-tail class distribution in nuScenes, we use the ground truth sampling in [60] to randomly paste objects into the current frame [60, 76]. We also adopt the class-balanced resampling and class-grouped heads in [76] to improve the average density of rare classes. We train the model for 20 epochs with the AdamW [34] optimizer using the one-cycle policy [16], with a max learning rate of 3e-3 following [66]. The training takes 2.5 days on 4 V100 GPUs with a batch size of 16 (4 frames per GPU).

During the testing, we set the output threshold to be 0.05 for the 2D detector and generate 50 virtual points for each 2D object in the scene. We use an output threshold of 0.1 for the 3D detector after performing non-maxima suppression with an IoU threshold of 0.2 following CenterPoint [66].

### 5.1 State-of-the-art Comparison

We first compare with state-of-the-art approaches on the nuScenes test set. We obtain all results on the public leaderboard by submitting our predictions to an online evaluation server. The submission uses a single MVP model without any ensemble or test-time augmentations. We compare to other methods under the same setting. Table 1 summarizes our results. On the nuScenes dataset, MVP achieves state-of-the-art results of 66.4 mAP and 70.5 NDS, outperforming the strong CenterPoint baseline by 8.4 mAP and 5.0 NDS. MVP shows consistent improvements across all object categories with significant 11 mAP accuracy boosts for small objects (+20.6 for Bicycle and +16.3 for motorcycle). These results clearly verify the effectiveness of our multi-modal virtual point approach.

### 5.2 Ablation Studies

**Comparison of 2D and 3D Detector.** We first validate the superior detection performance of the camera-based 2D detector compared to the Lidar-based 3D detector. Specifically, we use two state-of-the-art object detectors: CenterPoint [66] for Lidar-based 3D detection, and CenterNet [74] for image-based 2D detection. To compare the performance of detectors working in different modalities, we project the predicted 3D bounding boxes into the image space to get the corresponding 2D detections. The 2D CenterNet detector is trained with projected 2D boxes from ground truth 3D annotations. Table 2 summarizes the results over the whole nuScenes validation set. 2D CenterNet [74] significantly outperforms the CenterPoint model by 9.8 mAP (using 2D overlap). The improvements are larger for smaller objects with a 12.6 mAP improvement for objects of medium size and more than $3\times$ accuracy improvements for small objects (6.9 mAP vs. 1.6 mAP). See Figure 2 for a qualitative visualization of these two detectors' outputs. These results support our motivations for utilizing high-resolution image information to improve 3D detection models with sparse Lidar input.

Table 2: Quantitative comparison between state-of-the-art image-based 2D detector [74] and point cloud based 3D detector [66] on the nuScenes validation set measuring 2D detection accuracy (AP). The comparison use the COCO [31] style mean average precision with 2D IoU threshold between 0.5 and 0.95 in image coordinates. For 3D CenterPoint [66] detector, we project the predicted 3D bounding boxes into images to get the 2D detections. The results show that the 2D detector performs significantly better than Lidar-based 3D detector at localizing small or medium size objects due to high resolution camera input.

| Method | $AP_{small}$ | $AP_{medium}$ | $AP_{large}$ | AP |
|---|---|---|---|---|
| CenterPoint [66] | 1.6 | 11.7 | 34.5 | 22.7 |
| CenterNet [74] | **6.9** | **24.3** | **42.6** | **32.5** |

Table 3: Component analysis of our MVP model with VoxelNet [60, 75] and PointPillars [23] backbones on nuScenes validation set.

| Encoder | Baseline | Virtual Point | Split Voxelization | Two-stage | mAP↑ | NDS↑ |
|---|---|---|---|---|---|---|
| VoxelNet | ✓ | | | | 59.6 | 66.8 |
| | ✓ | | | ✓ | 60.5 | 67.4 |
| | ✓ | ✓ | | | 65.9 | 69.6 |
| | ✓ | ✓ | ✓ | | 66.0 | 70.0 |
| | ✓ | ✓ | ✓ | ✓ | **67.1** | **70.8** |
| PointPillars | ✓ | | | | 52.3 | 61.3 |
| | ✓ | ✓ | | | **62.7** | **66.1** |

**Component Analysis.**   Next, we ablate our contributions on the nuScenes validation set. We use the Lidar-only CenterPoint [74] model as our baseline. All hyperparameters and training procedures are the same between all baselines. We change inputs (MVP or regular points), voxelization, or an optional second stage. Table 3 shows the importance of each component of our MVP model. Simply augmenting the Lidar point cloud with multi-modal virtual points gives a 6.3 mAP and 10.4 mAP improvements for VoxelNet and PointPillars encoder, respectively. For the VoxelNet encoder, split voxelization gives another 0.4 NDS improvements due to the better modeling of features inside a voxel. Moreover, two-stage refinement with surface center features brings another 1.1 mAP and 0.8 NDS improvements over our first stage models with small overheads (1-2ms). The improvement of two-stage refinement is slightly larger with virtual points than without. This highlights the effectiveness of our virtual point method to create a finer local structure for better localization and regression using two-stage point-based detection.

**Performance Breakdown.**   To better understand the improvements of our MVP model, we show the performance comparisons on different subsets of the nuScenes validation set based on object distances to the ego-vehicle. We divide all ground truth annotations and predictions into three ranges: 0-15m, 15-30m, and 30-50m. The baselines include both the Lidar-only two-stage CenterPoint [66] model and the state-of-the-art multi-modal fusion method PointPainting [52]. We reimplement PointPainting using the same 2D detections, backbones, and tricks (including two-stage) as our MVP approach. The main difference to PointPainting [52] is our denser Lidar inputs with multimodal virtual points. Table 4 shows the results. Our MVP model outperforms the Lidar-only baseline by 6.6 mAP while achieving a significant 10.1 mAP improvement for faraway objects. Compared to PointPainting [52], our model achieve a 1.1 mAP improvement for faraway objects and performs comparatively for closer objects. This improvement comes from the dense and fine-grained 3D structure generated from our MVP framework. Our method makes better use of the higher dimensional RGB measurements than the simple point-wise semantic feature concatenation as used in prior works [48, 52].

**Robustness to 2D Detection**   We investigate the impact of 2D instance segmentation quality on the final 3D detection performance. With the same image network, we simulate the degradation of 2D segmentation performance with smaller input resolutions. We show the results in Table 5. Our

MVP model is robust to the quality of 2D instance segmentation. The 3D detection performance only decreases by 0.8 NDS with a 9 point worse instance segmentation inputs.

**Depth Estimation Accuracy**   We further quantify the depth estimation quality of our nearest neighbor-based depth interpolation algorithm. We choose objects with at least 15 lidar points and randomly mask out 80% of the points. We then generate virtual points from the projected locations of the masked out lidar points and compute the a bi-directional pointwise chamfer distance between virtual points and masked out real lidar points. Our nearest neighbor approach has bi-directional chamfer distance of 0.33 meter on the nuScenes validation set. We believe more advanced learning based approaches like  [18] and  [21] may further improve the depth completion and 3D detection performance.

**KITTI Results**   To test the generalization of our method, we add an experiment on the popular KITTI dataset [12]. For 2D detection, we use a pretrained MaskFormer [9] model to generate the instance segmentation masks and create 100 virtual points for each 2D object in the scene. For 3D detection, we use the popular PointPillars [23] detector with augmented point cloud inputs. All other parameters are the same as the default PointPillars model. As shown in Table 6, augmenting the Lidar point cloud with our multimodal virtual points gives a 0.5 mAP and 2.3 mAP for vehicle and cyclist class, respectively. We didn't notice an improvement for the pedestrian class, presumable due to inconsistent pedestrian definition between our image model (trained on COCO [32]) and the 3D detector. On COCO, people inside a vehicle or on top of a bike are all considered to be pedestrians while KITTI 3D detectors treat them as vehicle or cyclist.

Table 4: Comparisons between Lidar-only CenterPoint [66] method, fusion-based PointPainting [52] method (denoted as CenterPoint + Ours(w/o virtual)), and our multimodal virtual point method for detecting objects of different ranges. All three entries use the VoxelNet backbone. We split the nuScenes validation set into three subsets containing objects at different ranges.

| Method | nuScenes mAP | | | |
| --- | --- | --- | --- | --- |
| | 0-15m | 15-30m | 30-50m | Overall |
| CenterPoint [66] | 76.2 | 60.3 | 37.2 | 60.5 |
| CenterPoint + Ours(w/o virtual) | **78.2** | 67.4 | 46.2 | 66.5 |
| CenterPoint + Ours | 78.1 | **67.7** | **47.3** | **67.1** |

Table 5: Influence of 2D instance segmentation quality for the final 3D detection performance. We show the input resolution, 2D detection mAP, and 3D detection nuScenes detection score (NDS).

| Resolution | 2D mAP | NDS |
| --- | --- | --- |
| 900 | **43.3** | **70.0** |
| 640 | 39.5 | 69.6 |
| 480 | 34.2 | 69.2 |

Table 6: Comparison between Lidar-only PointPillars detector and our multimodal virtual point method for 3D detection on KITTI dataset. We show the 3D detection mean average precision for each class under the moderate difficulty level.

| Method | Car | Cyclist | Pedestrian |
| --- | --- | --- | --- |
| PointPillars [23] | 77.3 | 62.7 | **52.3** |
| PointsPillars+Ours | **77.8** | **65.0** | 50.5 |

## 6   Discussion and conclusions

We proposed a simple multi-modal virtual point approach for outdoor 3D object detection. The main innovation is a multi-modal virtual point generation algorithm that lifts RGB measurements into

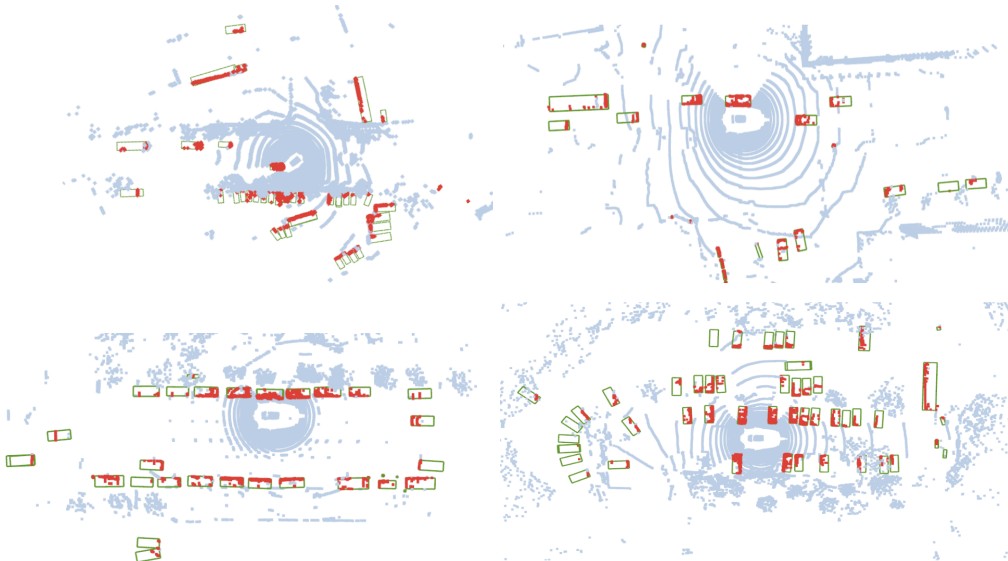

Figure 4: Example qualitative results of MVP on the nuScenes validation set. We show the raw point-cloud in blue, our detected objects in green bounding boxes, and Lidar points inside bounding boxes in red. Best viewed on screen.

3D virtual points using close-by measurements of a Lidar sensor. Our MVP framework generates high-resolution 3D point clouds near target objects and enables more accurate localization and regression, especially for small and faraway objects. The model significantly improves the strong Lidar-only CenterPoint detector and sets a new state-of-the-art on the nuScenes benchmark. Our framework seamlessly integrates into any current or future 3D detection algorithms.

There are still certain limitations with the current approach. Firstly, we assume that virtual points have the same depth as close-by Lidar measurements. This may not hold in the real world. Objects like cars don't have a planar shape vertical to the ground plane. In the future, we plan to apply learning-based methods [50, 70] to infer the detailed 3D shape and pose from both Lidar measurements and image features. Secondly, our current two-stage refinement modules only use features from the bird-eye view which may not take full advantage of the high-resolution virtual points generated from our algorithm. We believe point or voxel-based two-stage 3D detectors like PVRCNN [45] and M3Detr [15] may give more significant improvements. Finally, the point-based abstraction connecting 2D and 3D detection may introduce too large of a bottleneck to transmit information from 2D to 3D. For example, no pose information is contained in our current position + class based MVP features.

Overall, we believe future methods for scalable 3D perception can benefit from the interplay of camera and Lidar sensor inputs via dense semantic virtual points.

**Societal Impacts.**    First and foremost better 3D detection and tracking will lead to safer autonomous vehicles. However, in the short term, it may lead to earlier adoption of potentially not-yet safe autonomous vehicles, and misleading error rates in 3D detection may lead to real-world accidents. Fusing multiple modalities may also increase the iteration cycle and safety testing requirements of autonomous vehicles, as different modalities clearly adapt differently to changes in weather, geographic locations, or even day-night cycles. A low sun may uniquely distract an RGB sensor, and hence unnecessarily distract a 3D detector through MVPs.

Furthermore, increasing the reliance of autonomous vehicles on color sensors introduces privacy issues. While most human beings look indistinguishable in 3D Lidar measurements, they are clearly identifiable in color images. In the wrong hands, this additional data may be used for mass surveillance.

**Acknowledgement**    We thank the anonymous reviewers for the constructive comments. This material is based upon work supported by the National Science Foundation under Grant No. IIS-1845485 and IIS-2006820.

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
