# OpenReview forum: "Multimodal Virtual Point 3D Detection"
_NeurIPS.cc/2021/Conference — NeurIPS 2021 Poster_

### Official Review · Reviewer_r8dG · 2021-07-11

**Rating:** 6
**Confidence:** 5

**Summary:**

The paper argues that despite being accurate in 3D position, point cloud is more sparse than rgb camera. Thus, the paper proposes to generate virtual points by lifting 2D pixels to 3D space, and helps the detection of far-away objects. Experiments on Nuscenes dataset validate the effectiveness of the proposed method.

**Limitations And Societal Impact:**

The authors pointed out the limitations and potential genative societal impact of the work.



**Main Review:**

Strengths:

1. The proposed method is well-motivated and the objective is clear.

2. The paper is generally well-written and easy to follow

3. Potentially the proposed point augmentation algorithm can be used on a lot more AD datasets.

Major Weaknesses:

1. The author misses the related work on point cloud augmentation.

2. The paper lacks theoretical novelty. The proposed point cloud augmentation seems to be a combination of Frustum PointNet and CenterNet, with a point projection and a nearest neighbor search modules. The contribution leans more to the experiment and network tuning part.

3. Experiments only done on Nuscenes. The proposed pipeline is really neat and straightforward, so it should be easy to test on multiple datasets. Results on at least one more public benchmark (E.g. KITTI, Waymo) will make the statement of the paper more convincing.

Minor Weaknesses:

1. The authors make a statement that 2D detectors have generally higher 2D detection than 3D detectors. However, on KITTI 2D detection benchmark, the highest numbers are almost all achieved by 3D detectors like BANet, SE-SSD, PV-RCNN, etc. Thus, the statement made here is not quite accurate.

2. The author uses nearest neighbor when doing the inverse projection from image pixel to 3D point. However, this is a very coarse approximation, because not all planes are perpendicular to the z axis. A more accurate approximation would be: To find the K nearest neighbor points, fit a local plane, and then find the intersection point between the camera lifted ray and that plane. In this way, the virtual points would be more smooth in space as they lie on the local plane.

**Time Spent Reviewing:**

3 hrs

---

> ### Author Response · Authors · 2021-08-10
> **Reply to Reviewer r8dG**
>
> We thank the reviewer for the careful review. We are encouraged that the reviewer found our approach to be well-motivated, neat and straightforward, and potentially can be used in a lot more AD datasets. We address the remaining concerns below.
>
> **Q1**: Missing related works on point cloud augmentation.
>
> **A1**: In the related work section, we think the closest methods to our approach are [37] [51] which generates virtual lidar points from monocular or stereo images for camera-based 3D detection. We will add more discussion about point upsampling methods like [6,7,8]. We will also ensure to incorporate any related works that the reviewers deem appropriate in the final version.
>
> **Q2**: The paper lacks theoretical novelty. The proposed point cloud augmentation seems to be a combination of Frustum PointNet and CenterNet, with a point projection and a nearest neighbor search modules
>
> **A2**: We believe our framework has a significant difference from Frustum PointNet + CenterNet. Frustum PointNet uses 2D detections to crop the point cloud and run 3D detections on each **individual** crop. In contrast, we do not crop the point cloud, but add virtual points to the point cloud and run 3D detection on the augmented point cloud as a whole using standard 3D detectors. At a high level, Frustum PointNet is more moving point from 3D to 2D, and we are moving pixels from 2D to 3D. Frustum PointNet could not recover from missing 2D detections, while our framework can detect new objects that are missed both in the original 2D or 3D detectors.
>
> Empirically, our MVP approach also tends to be both **robust** and **general**.
>
> Specifically, as shown in the new experiments requested by Reviewer FiPr, our MVP approach is robust to the instance segmentation quality. Concretely, we use the same image network with different input resolutions to analyze the impact of the quality of 2D instance segmentation on downstream 3D object detection. The results are below.
>
> |Resolution | 2D mAP | NDS |
> | --------------| ------------| -------|
> |900 (original one in paper)   |  43.3 |  70 |
> |640  | 39.5 | 69.6 |
> |480 | 34.2 | 69.2 |
>
> The drop of 3D accuracy is only 0.8 percent with a 9 point worse instance segmentation input.
>
> To further verify that our framework is general and can be applied to more 2D and 3D detectors, we tried another 2D instance segmentation network Cascade Mask-RCNN and another 3D detector PointPillar [4] (in addition to the CenterNet2 2D detector and CenterPoint-VoxelNet 3D detector in the original submission). Replacing the CenterNet2 detector with Cascade Mask-RCNN gives similar performance improvement on the nuScenes dataset. Combining our MVP approach with PointPillar yields a significant **10.4** mAP improvement (detailed results are below).
>
> | Method | mAP | NDS |
> |---------| ---------| ---------|
> | PointPillar |  52.3  |      61.3     |
> | PointPillar+MVP (ours) | **62.7** | **66.1** |
>
> For these reasons, we believe our paper is a conceptually novel and solid contribution to multi-modal 3D object detection.
>
> **Q3**: More results on one more public dataset?
>
> Thank you for the suggestion. We experimented with KITTI during the rebuttal period. We use the PointPillar [4] encoder with a pretrained instance segmentation model from [5]. Results are listed below. We show the comparison of 3D mAP under the moderate split of the KITTI validation set.
>
> | Method | Car | Cyclist |
> |------------|------|-----------|
> | PointPillar | 77.3 | 62.7 |
> | PointPillar+MVP (ours) | **77.6** | **65.0** |
>
> Compared to the Lidar-only PointPillar baseline, we notice a 0.3 improvement for the car category and 2.3 mAP for the cyclist class.
>
> **Q4**: The author's statement that 2D detectors get higher 2D accuracy than 3D detectors is not quite accurate. On KITTI, highest numbers are all achieved by 3D detectors like BANet, SE-SSD, PV-RCNN, etc.
>
> **A4**: We have verified our claims on nuScenes and we show that a standard 2D detector gets **9.8** mAP higher 2D detection accuracy than the **state-of-the-art** Lidar-based 3D detector (see table 2). We agree with the reviewer that most top methods on the KITTI 2D benchmark are Lidar only 3D detectors (e.g. BANet, SE-SSD, PV-RCNN). However, we also note that the top-performing methods on KITTI often include a 2D detector in the framework. For example, CLOCS_PVCas [1] makes heavy use of a Cascade RCNN 2D detector to locate 2D bounding boxes and achieves the best 95.96 car detection mAP (compared to the 95.61 mAP for BANet, 95.60 for SE-SSD, and 94.7 for PV-RCNN). Similarly, Frustum-PointNet [2] uses an SSD-based 2D detector [3] and achieves **80.13** pedestrian detection mAP compared to the **58.37** for PV-RCNN (the other two models didn’t submit pedestrian results.)
>
> **Q5**: Nearest neighbor is a coarse approximation for depth inference. What about using the k nearest neighbor approach? The generated virtual points will be more smooth.
>
> **A5**
> : Thank you for the suggestion. We tried this and the results are below:
>
> | K | 3D NDS |
> |-----|---------------|
> | 1 (Ours) |      **70.0** |
> | 3 |       67.8        |
> | 5 |       66.7        |
>
> One nearest neighbor is simple and fast, and achieves the best 3D detection performance. The tradeoff is simple: more neighbors will yield a more detailed geometry, but introduce more false positives (neighbors). In our experiments, the wrongly chosen neighbors led to a larger loss in performance than the smooth geometry was able to recover.
>
> [1] Pang et al. CLOCs: Camera-LiDAR object candidates fusion for 3D object detection, IROS 2020
>
> [2] Qi et al. Frustum pointnets for 3d object detection from rgb-d data, CVPR 2018
>
> [3] Liu et al. Ssd: Single shot multibox detector, ECCV 2016
>
> [4] Lang et al. PointPillars: Fast Encoders for Object Detection from Point Clouds, CVPR 2020
>
> [5] Cheng et al. Per-Pixel Classification is Not All You Need for Semantic Segmentation, arXiv 2021
>
> [6] Yu et al. Pu-net: Point cloud upsampling network, CVPR 2018
>
> [7] Wang et al. Patch-based progressive 3d point set upsampling, CVPR 2019
>
> [8] Li et al. Pu-gan: a point cloud upsampling adversarial network, CVPR 2019
>
> [37]  Qian et al. End-to-end pseudo-lidar for image-based 3d object detection, CVPR 2020
>
> [51] Yan et al. Pseudo-lidar from visual depth estimation: Bridging the gap in 3d object detection for autonomous driving, CVPR 2019

---

> > ### Comment · Reviewer_r8dG · 2021-08-28
> > **Thanks for the response**
> >
> > 1. Thanks authors for providing the response. The response addresses most of my concerns about generalization, model design and missed literature. In light this, I would like to update my rating from 4 to 6.
> >
> > 2. I suggest that the authors include a more detailed experimental part of KITTI dataset. Also, please update the necessary experiments in the final manuscript.

---

### Official Review · Reviewer_PYBU · 2021-07-16

**Rating:** 7
**Confidence:** 4

**Summary:**

This paper proposes a new method (MVP) to use multimodal (RGB+LiDAR) sensor data in 3D perception. MVP is simple in principle: compute the 2D segmentation mask, project LiDAR points on the mask, and generate the virtual points by sampling the points in the mask and attaching them with the closest LiDAR depth. Though simple, the authors show empirically the proposed method can bring performance gain and achieve SOTA results on the nuScenes 3D detection task.

**Limitations And Societal Impact:**

The authors have adequately addressed the limitations and potential negative societal impact of their work.

**Main Review:**

Pros:
1. The paper is well written and easy to read.
2. I personally like that the proposed method is simple and light while effective. The method is sound and reasonable to me.
3. Good experiment results and ablation study support the proposed method.

Cons:
1. The experiments are solely on nuScenes dataset, whose LiDAR is extremely sparse. LiDARs are getting denser and denser nowadays, thus I wonder if the proposed method can still bring gain on datasets with denser LiDARs, e.g. KITTI and Waymo Open Dataset? This can also showcase the generalizability of the proposed method.
2. Could you show more qualitative visualizations on both success and failure cases of the virtual points? As the authors have mentioned in L259, assigning the depth of virtual points to the close-by LiDAR measurement might not be a good estimate in the real world. Though the effectiveness of the MVP is validated by object detection tasks, I would suggest the authors conduct some experiments to show how well the estimate (i.e. take the depth of close-by LiDAR as virtual point depth) can be, for example, take objects with dense LiDAR, randomly mask out most of the LiDAR points, generate virtual points and quantize the closeness by computing the Chamfer distance.

=======

Post Rebuttal Comments:
Thanks for the authors' effort in putting up the response. The responses address my concerns, though I would suggest including detailed results on the KITTI dataset (include car, pedestrian, and cyclist results.). I decided to keep my original rating.

**Time Spent Reviewing:**

1.5

---

> ### Author Response · Authors · 2021-08-10
> **Reply to Reviewer PYBU**
>
> We thank the reviewer for the positive feedback and constructive suggestions. We answer the questions below.
>
> **Q1**: More results on KITTI / Waymo to check if the method works with denser Lidar?
>
> **A1**: Thanks for the suggestion. We added a KITTI experiment during the rebuttal. We use the PointPillar [1] encoder with a pretrained instance segmentation model from [2]. Results are listed below. We show the comparison of 3D mAP under the moderate split of the KITTI validation set.
>
> | Method | Car | Cyclist |
> |------------|------|-----------|
> | PointPillar [1] | 77.3 | 62.7 |
> | PointPillar+MVP(ours) | **77.6** | **65.0** |
>
> Compared to the Lidar-only PointPillar baseline, we notice a 0.3 improvement for the car category and 2.3 mAP for the cyclist class. These results show that our model can still improve 3D detection performance with denser Lidar point clouds.
>
> **Q2**: More visualization of the success and failure cases.
>
> **A2**: That's a good point and we will add more visualization to the final paper. The success modes include detailed 3D structure for faraway objects which enables better classification, localization, and orientation estimation (see the pedestrian in Figure 1). Failure modes mostly come from incomplete shape or outlier virtual points due to the 2D segmentation error or object occlusions.
>
> **Q3**: Quantify the depth prediction accuracy of the proposed approach using chamfer distance.
>
> **A3**: Thanks for the suggestion. We choose objects with at least 15 lidar points and randomly mask out 80% of the points. We then generate virtual points from the projected locations of the masked out lidar points and compute the bi-directional pointwise chamfer distance between virtual points and masked out real lidar points. We found that our nearest neighbor approach works reasonably well with a bi-directional chamfer distance of 0.33 meter on the nuScenes validation set. We will add this result to the paper.
>
> [1] Lang et al. PointPillars: Fast Encoders for Object Detection from Point Clouds CVPR 2020
>
> [2] Cheng et al. Per-Pixel Classification is Not All You Need for Semantic Segmentation, arXiv 2021

---

### Official Review · Reviewer_FiPr · 2021-07-16

**Rating:** 4
**Confidence:** 4

**Summary:**

Considering the sparse nature of point clouds within distant objects, this paper proposes to generate dense 3D virtual points by nearest interpolation from sparse point based on the instance mask provided by the 2D network to augment the sparse 3D point-cloud. In this way, the point cloud within the distant object will be denser, so that the downstream 3D object detector can localize distant objects better. The method in this paper makes good use of the high resolution of the 2D network for distant objects.

**Limitations And Societal Impact:**

No obvious limitations and potential negative societal impact

**Main Review:**

- The paper uses the advantages of 2D image detection on distant objects to search for possible instance locations in 2D. It then extracts the sparse point clouds inside these instances for densification, making full use of the advantages of 2D information and 3D information. It achieves good results and a certain degree of innovation. The design of the method itself is simple, and the authors have explained the proposed method very clear.
- My concern is that adding a 2-stage instance segmentation network will bring a significant burden to the whole network. If this part of the computation is added to a normal 3D detection network through network scaling, will it improve performance at the same level? As a result, this is not a fair comparison to the non-ensembled methods as the amount of computation is not consistent.
- Another question is that compared to PointPainting, which requires a segmentation network, virtual points need a more complex 2D instance segmentation network. Will this weaken the generalization ability of the model? It's better to verify on different data sets.
- Need to add the analysis of the impact of the quality of 2D instance segmentation on downstream 3D object detection because the quality of 2D instance segmentation cannot be guaranteed under different conditions.
- The authors claim that the framework is a plug-and-play module to any existing or new 2D or 3D detectors, but the paper only verifies one type of 2D/3D detector. It's better to perform experimental verification on different 2D/3D detectors to show its generalization.
- If I understand correctly, the only difference between CenterPoint + PointPainting and CenterPoint + Ours in Table 4 is that virtual points are added, am I right? It needs to be clarified, or it can be modified to CenterPoint + Ours(w/o virtual points) / CenterPoint + Ours. CenterPoint + PointPainting is a little confusing.
- Better add a gain comparison for the category with small size.

**Time Spent Reviewing:**

3

---

> ### Author Response · Authors · 2021-08-10
> **Reply to Reviewer FiPr**
>
> We thank the reviewer for the detailed reviews and constructive comments. We are encouraged that the reviewer found our method to be simple, innovative, and achieves good results. The two main concerns are: 1) instance segmentation model used in our network brings significant overhead (Q1). 2) The instance segmentation network is more complex and non-standard compared to the semantic segmentation in PointPainting and may weaken generalization (Q2). We address them first and respond to the remaining questions and suggestions.
>
> **Q1**: The 2-stage instance segmentation brings **significant** burden to the whole network so the comparison with the Lidar-only approach is not fair due to inconsistent computation.
>
> **A1**: We believe this is a misunderstanding. Our 2-stage instance segmentation runs at **40** FPS without any optimization (**60-80** FPS with FP16+tensorRT). The Lidar backbone runs at 10 FPS. So the image branch adds little latency overhead to the whole network. To verify that the improvements do not just come from this additional computation, we compare with an ensemble version of the state-of-the-art Lidar-only CenterPoint model [1]. Specifically, CenterPoint introduces a double flip testing strategy where the model takes four rotated copies of the original point clouds and averages the outputs. This flip testing increases the runtime to 4X (adding the latency by about 300ms) and is the preferred scaling technique by challenge entries(e.g., CenterPoint [1], PointAugmenting [6]). The results are below.
>
> | Method | NDS |
> |--------|------|
> | CenterPoint | 66.8  |
> | CenterPoint + Flip Testing | 68.5 |
> | CenterPoint + MVP (ours) | **70.0** |
>
> We can see that our MVP model is still 1.5 NDS better than the ensembled CenterPoint model.
>
> **Q2**: The proposed approach requires a more complex 2D instance segmentation network while PointPainting only needs a semantic segmentation network. Will this weaken the generalization ability? The author should verify on different datasets.
>
> **A2**: We use an instance segmentation model following the common practice of fusion-based approaches on nuScenes [1] [2]. Notably, while the original PointPainting paper uses semantic segmentation to paint point clouds, popular follow-up works including two challenge winners of the nuScenes benchmark [1] [2] all implement PointPainting using instance segmentation as this turns out to be faster in general (40 FPS for our CenterNet2 instance segmentation model vs. 3 FPS for the deeplab v3+ model used in PointPainting [3]) and achieves similar performance improvement [1][3].
>
> Additionally, to the best of our knowledge, there is no evidence showing that instance segmentation generalizes worse than semantic segmentation and empirically we notice the improvements are similar across the validation and the hidden testing set.
>
> During the rebuttal, we further add an experiment on the KITTI dataset using the PointPillar [4] encoder with a pretrained instance segmentation model from [5]. Results are listed below. We show the comparison of 3D mAP under the moderate split of the KITTI validation set.
>
> | Method | Car | Cyclist |
> |------------|------|-----------|
> | PointPillar [4] | 77.3 | 62.7 |
> | MVP (ours) | **77.6** | **65.0** |
>
> Compared to the Lidar-only PointPillar baseline, we notice a 0.3 improvement for the car category and 2.3 mAP for the cyclist class. We believe our results on nuScenes and KITTI verify the generalization ability of the proposed approach.
>
> **Q3**: Need to add the analysis of the impact of the quality of 2D instance segmentation on downstream 3D object detection because the quality of 2D instance segmentation cannot be guaranteed under different conditions.
>
> **A3**: Thanks for the suggestion! We use the same image network with different input resolutions to analyze the impact of the quality of 2D instance segmentation on downstream 3D object detection. The results are below.
>
> |Resolution | 2D mAP | NDS |
> | --------------| ------------| -------|
> |900 (original one in paper)   |  43.3 |  70 |
> |640  | 39.5 | 69.6 |
> |480 | 34.2 | 69.2 |
>
> We can see that our model is robust to the quality of 2D instance segmentation. The 3D detection performance only decreases by 0.8 NDS with a 9 point worse instance segmentation inputs.
>
> **Q4**: The authors should try different 2D/3D detectors to verify that the framework is general.
>
> **A4**: Thanks for the suggestion. We tried Cascade Mask-RCNN with ResNet50 backbone and 1600x900 input resolution in our early experiments. The overall model performs similarly to our current approach with the CenterNet2 object detector with DLA-34 [7] backbone and 900x608 image resolution but it is slower (10 FPS vs. 40 FPS).
>
> During the rebuttal, we also tested another popular 3D detector PointPillar (in addition to the VoxelNet in the submission). The results are below.
>
> | Method | mAP | NDS |
> |---------| ---------| ---------|
> | PointPillar |  52.3  |      61.3     |
> | PointPillar+MVP (ours) | **62.7** | **66.1** |
>
>
> Our MVP model achieves **10.4** mAP improvement on nuScenes validation compared to the PointPillar baseline. We believe these results together with the submission results on 2 popular 2D detectors (CenterNet and Mask-RCNN) and 2 popular 3D detectors (VoxelNet and PointPillar) verify that the proposed method is general.
>
> **Q5**: Confusion about the name of entries in Table 4. Maybe rename it to CenterPoint + Ours(w/o virtual points) / CenterPoint + Ours.
>
> **A5**: Thank you for the suggestion. Your interpretation is right. In Table 4, the baseline CenterPoint+PointPainting uses the same 2D instance segmentation network as our approach to annotate **real** lidar points with semantic labels. We will rename the entries following the reviewer’s suggestion.
>
> **Q6**: Add a gain comparison for the category with small size
>
> **A6**: Thanks for the suggestion. We show the comparison table below and we will add it to the final version. Compared to the CenterPoint baseline, we get a significant **9.9** mAP improvement for categories with small size with a notable **13.7** mAP for the motorcycle category and **19.7** for the bicycle class.
>
> | Method | Pedestrian | Motor | Bicycle | Traffic Cone | Barrier |  Overall |
> |---------| ---------| ---------|-------|-----------|-----------------|-----------|
> | CenterPoint |    85.1     |  58.9    |   43.4      |  69.7    |  68.6   |   65.1    |
> | CenterPoint + MVP (ours) | **89.4** | **72.6** | **63.1** | **79.5** | **70.5** | **75.0** |
>
>
> We hope most concerns are resolved in this rebuttal and the reviewer may consider adjusting the assessment. Please let us know for any further questions.
>
> [1] Yin et al. Center-based 3D Object Detection and Tracking, CVPR 2021
>
> [2] Xu et al. FusionPainting: Multimodal Fusion with Adaptive Attention for 3D Object Detection, ITSC 2021
>
> [3] Vora et al. PointPainting: Sequential Fusion for 3D Object Detection CVPR 2019
>
> [4] Lang et al. PointPillars: Fast Encoders for Object Detection from Point Clouds CVPR 2020
>
> [5] Cheng et al. Per-Pixel Classification is Not All You Need for Semantic Segmentation, arXiv 2021
>
> [6] Wang et al. PointAugmenting: Cross-Modal Augmentation for 3D Object Detection, CVPR 2021
>
> [7] Yu et al. Deep layer aggregation, CVPR 2018

---

### Decision · Program_Chairs · 2021-09-27

**Decision:**

Accept (Poster)

**Comment:**

This paper approaches the problem off 3D object classification on multimodal sparse LIDAR + dense RGB data by suggesting to use 2D object and region detectors on RGB images, dense 3D “viirtual” point generation by picking 2D points from detected instances nearby sparse 3D points, assuming that the depth of these sampled neighbors is the same as the reference, and then building “dense” 3D “virtual” point clouds to be classified. The technique is evaluated on nuScenes, where it achieves state-of-the-art 3D detection.

Reviewers praised the simplicity of the method and clarity of the paper, as well as the good experimental results. Reviewers had many issues with the paper: computational requirements of the method (FiPr), generalisability to KITTI (FiPr, PYBU, r8dG), flexibility with other detectors (FiPr), performance on small objects (FiPr), literature on point cloud augmentation (r8dG). All these questions were addressed in the rebuttal with additional experimental results and explanations.

One reviewer gave a score of 7 and another one off 6. One reviewer (FiPr) gave a scores of 4 but did not update their reviews after considerable rebuttal and additional experimental work by the authors. While waiting on reviewers FiPr, I am therefore willing to challenge the reviewers and promote this paper to an acceptance.